# Integration of Multiomic Data to Characterize the Influence of Milk Fat Composition on *Cantal*-Type Cheese Microbiota

**DOI:** 10.3390/microorganisms10020334

**Published:** 2022-02-01

**Authors:** Marie Frétin, Amaury Gérard, Anne Ferlay, Bruno Martin, Solange Buchin, Sébastien Theil, Etienne Rifa, Valentin Loux, Olivier Rué, Christophe Chassard, Céline Delbès

**Affiliations:** 1UMR 0545 Fromage, Université Clermont Auvergne, INRAE, VetAgro Sup, 20 Côte de Reyne, F-15000 Aurillac, France; fretin.marie7@gmail.com (M.F.); amaury.gerard@uliege.be (A.G.); sebastien.theil@inrae.fr (S.T.); christophe.chassard@inrae.fr (C.C.); 2Laboratory of Quality and Safety of Agrofood Products, Gembloux Agro-Bio Tech, University of Liège, 2 Passage des Déportés, 5030 Gembloux, Belgium; 3UMR 1213 Herbivores, Université Clermont Auvergne, INRAE, VetAgro Sup, 63122 Saint-Genès-Champanelle, France; anne.ferlay@inrae.fr (A.F.); bruno.martin@inrae.fr (B.M.);; 4URTAL, INRAE, F-39800 Poligny, France; solange.buchin@inrae.fr; 5Toulouse Biotechnology Institute (TBI), Université de Toulouse, CNRS, INRAE, INSA, F-31077 Toulouse, France; etienne.rifa@inrae.fr; 6Plateforme Genome et Transcriptome (GeT), Genopole Toulouse, F-31077 Toulouse, France; 7MIGALE Bioinformatics Facility, Université Paris-Saclay, INRAE, BioinfOmics, F-78350 Jouy-en-Josas, France; olivier.rue@inrae.fr (O.R.); valentin.loux@inrae.fr (V.L.); 8MaIAGE, Université Paris-Saclay, INRAE, F-78350 Jouy-en-Josas, France;

**Keywords:** cheese microbiota, bacteria, fungi, metabarcoding, lipidomics, volatolomics, dairy cow, pasture

## Abstract

A previous study identified differences in rind aspects between *Cantal*-type cheeses manufactured from the same skimmed milk, supplemented with cream derived either from pasture-raised cows (P) or from cows fed with maize silage (M). Using an integrated analysis of multiomic data, the present study aimed at investigating potential correlations between cream origin and metagenomic, lipidomic and volatolomic profiles of these *Cantal* cheeses. Fungal and bacterial communities of cheese cores and rinds were characterized using DNA metabarcoding at different ripening times. Lipidome and volatolome were obtained from the previous study at the end of ripening. Rind microbial communities, especially fungal communities, were influenced by cream origin. Among bacteria, *Brachybacterium* were more abundant in P-derived cheeses than in M-derived cheeses after 90 and 150 days of ripening. *Sporendonema casei*, a yeast added as a ripening starter during *Cantal* manufacture, which contributes to rind typical aspect, had a lower relative abundance in P-derived cheeses after 150 days of ripening. Relative abundance of this fungus was highly negatively correlated with concentrations of C18 polyunsaturated fatty acids and to concentrations of particular volatile organic compounds, including 1-pentanol and 3-methyl-2-pentanol. Overall, these results evidenced original interactions between milk fat composition and the development of fungal communities in cheeses.

## 1. Introduction

Both biotic and abiotic factors govern the formation of cheese microbiota. Biotic factors include inoculated microorganisms, i.e., starters used for manufacture or ripening, and endogenous microbiota [1]. The assembly of cheese microbiota is influenced by abiotic factors related to milk primary production conditions, such as the animal feeding system [2]. Workers involved in cheese manufacture, as well as tools, surfaces and ingredients, e.g., salt and water, contribute to its establishment [3]. The role of microorganisms in the development of cheese sensorial characteristics is well documented. Indeed, in combination with enzymes naturally found in raw milk, cheese microbiota are responsible for sugar fermentation, proteolysis and lipolysis, releasing volatile organic compounds (VOCs) associated with flavors. Consequently, raw milk cheeses show greater sensorial complexity in comparison to similar varieties manufactured from pasteurized milk [4]. Notably, during a study performed on *Salers* cheeses, [5] observed that sensorial characteristics differed between cheeses manufactured from heat-treated milk inoculated with various microbial consortia, supporting the hypothesis that milk/cheese microbial community structure influences cheese sensorial profiles.

Besides the influence of milk heat treatment, the sensorial quality of cheese also depends on milk biochemical composition, including milk fat concentration and composition. The role of lipolysis and free fatty acids (FFA) in the development of cheese flavor is essential, as most flavoring molecules are more liposoluble than hydrophilic. Animal diet is an easy way to modulate milk lipid content and profile. Grazing is, for instance, associated with considerable variations in milk fatty acid (FA) composition in comparison to diets based on maize silage and concentrates [6].

In *Morbier* cheese, similar sensory profiles were reported between cheeses made from the milk of pasture-raised cows and from the milk of cows fed with hay [7]. The same conclusion was reported for *Cantal* cheeses manufactured from the milk of grazing cows and from cows fed with maize silage [4]. In contrast, the effect of milk FA profiles on texture and rind aspect has already been demonstrated in *Cantal* cheeses made from pasture milk. The latter cheeses had a smoother texture than those manufactured from milk produced by cows fed with maize silage [4]. Rinds of *Cantal* cheeses from raw pasture milk were thinner and with smaller spots in comparison to those made from milk of cows receiving a maize-based diet [4]. These observations echo the feedbacks of cheese-makers who reported occasional difficulties in controlling the formation of the rind without any clearly identified explanatory factors. Frétin et al. [4] hypothesized that differences in the rind aspect could be due to an influence of milk FA profiles on the development of rind microbiota.

Extensive literature is available regarding rind bacterial community structure, including smear-ripened soft cheeses (*Maroilles* and *Munster* [8]), mold-ripened soft cheeses (*Saint-Marcellin* [8]) and uncooked pressed cheeses (*Cantal* [2] and *Saint-Nectaire* [8]). Nevertheless, data on rind fungal populations are less abundant, especially using next-generation sequencing (NGS) techniques [8,9,10,11].

In *Cantal* cheese, typical rind development involves the addition of fungal species as ripening starters, namely *Debaryomyces hansenii*, *Penicillium fuscoglaucum* and *Sporendonema casei*. The objective of this study was to investigate the influence of milk FA composition on the dynamics of bacterial and fungal communities in the core and rind of *Cantal*-type cheeses during ripening using metabarcoding. In order to identify potential explanatory factors for the effects of fat, an integrative analysis of omics data was carried out to explore relationships between the composition of the microbiota and concentrations of FA and volatile compounds in ripened cheeses.

## 2. Materials and Methods

### 2.1. Dairy Herd Management

The experiment was conducted at the experimental facility of INRAE Herbipôle in Marcenat (45°15′ N, 2°55′ E; 1135–1215 m above sea level, Cantal, France). Two groups of dairy cows were fed with different diets in order to modulate milk FA composition, as previously described by Frétin et al. [4]. Briefly, the pasture (P) group was composed of 27 cows (14 Holstein and 13 Montbéliarde, i.e., dairy breeds most commonly found in farms from the concerned geographical area (Auvergne Region)) grazed on mountain grassland regrowths (69% grasses, 21% forbs and 10% legumes), with a daily supplement of 1.4 kg (DM) of concentrate and 0.2 kg of minerals and vitamins premix. Twelve cows (6 Holstein and 6 Montbéliarde) formed the maize (M) group and were kept indoors and fed daily with 15 kg (DM) of maize silage, 2.5 kg (DM) of straw, and 5.6 kg (DM) of concentrate distributed twice a day (forage to concentrate ratio: 75.8:24.2). The animals were divided into these 2 groups based on breed, parity (41 and 42% of primiparous cows in P and M groups, respectively) and lactation stage (228 ± 121 and 223 ± 59 DIM (day in milk) in the P and M groups, respectively). All cows were milked twice daily in the same herringbone milking parlor.

### 2.2. Cheese Manufacture

This study is based on the same cheese batches as those previously considered by Frétin et al. [4]. Briefly, on each production day, all cheeses were produced from the same skimmed milk obtained from the P group cows, avoiding bias introduced by initial differences in microbiota and biochemical composition between milks—except that from cream fat matter. Skimmed milk was supplemented either with P- or M-derived pasteurized cream to obtain the same final concentration of fat (39 g/L) in each vat (Figure 1). It was thus hypothesized that, in this experiment, microbiota mainly originated from P skimmed milk and that the addition of pasteurized cream did not introduce significant variations in milk microbiota. For each cheese manufacturer, P and M morning milks were cooled down to 4 °C, and pooled with the following evenings’ milk for overnight storage at 4 °C, before being transported in airtight cans to the cheese-making pilot-scale facility (INRAE, UMR545, Aurillac). After milk skimming, creams were pasteurized at 70 °C for 30 min. Cheese manufacturing was repeated three times over one week with raw P skimmed milk and three times on the following week with pasteurized P skimmed milk (78 °C for 10 s). For each cheese manufacturing cycle, raw or pasteurized P skimmed milk was supplemented with P or M pasteurized cream to reach a fat concentration of 38 g/L. Small-size *Cantal* cheeses (10 kg) were manufactured simultaneously in distinct vats from 110 L of P- or M-derived milk. Overall, all types of cheeses were manufactured in triplicate, for a total of 12 production cycles performed over a period of two consecutive weeks. Cheese-making was performed according to Frétin et al. [2,12]. Briefly, milk was inoculated with Flora Danica (i.e., *Lactococcus lactis* ssp. *lactis*, *Lactococcus lactis* ssp. *cremoris*, *Lactococcus lactis* ssp. *lactis* biovar *diacetylactis* and *Leuconostoc mesenteroides* ssp. *cremoris*; CHR Hansen, Saint-Germain-lès-Arpajon, France) at 0.05 g/100 kg, and Monilev (i.e., *D. hansenii* and *S. casei*; Laboratoire Interprofessionnel de Production (LIP), Aurillac, France) and Penbac (*Brachybacterium tyrofermentans*, *Brevibacterium linens* and *P. fuscoglaucum*; LIP, Aurillac, France) as ripening starters. All cheeses considered during this work were manufactured using the same batches of starter cultures in order to avoid any bias. Rennet Fabre 520 (LCP Food Ingredients, Prat, France) was added at a concentration of 0.33 g/kg.

### 2.3. DNA Metabarcoding

Total DNA from raw skimmed milks added with either P- or M-derived pasteurized cream (10 mL), from cheese rinds (500 mg) at 30, 90 and 150 days of ripening (i.e., D30, D90 and D150, corresponding to the onset, an intermediate point in the development of the expected cheese rind and to an average ripening duration for Cantal cheese, respectively), from cheese cores (500 mg) at the first day of ripening (i.e., D3) and at D150 was extracted using phenol-chloroform extraction, as previously detailed by Frétin et al. [2]. These sampling times were selected based on a previous study performed on a similar cheese variety [5]. Regarding bacteria, V3-V4 regions of 16S rRNA gene (~510 bp) were amplified using primers PCR1F_460 (5′-TACGGRAGGCAGCAG-3′) and PCR1R_460 (5′-TTACCAGGGTATCTAATCCT-3′). Each PCR was carried out with a final volume of 50 μL, corresponding to 1 μL of dNTP mixture (10 mM), 1.25 μL of each primer (20 μM), 5 μL of 10X buffer and 0.5 μL of 5 U MolTaq DNA polymerase (Médiane Diagnostics, Plaisir, France). Fungal communities of cheese rinds were also explored through amplification of ITS2 region (~350 bp) using the primers ITS3f (5′-GCATCGATGAAGAACGCAGC-3′) and ITS4_KYO1 (5′-TCCTCCGCTTWTTGWTWTGC-3′), as detailed by Bokulich et al. [13]. Each PCR was carried out in a final volume of 25 μL, including 0.5 μL of dNTP mixture (10 mM), 0.75 μL of each primer (20 μM), 2.5 μL of 10X buffer, and 0.25 μL of 5 U MolTaq DNA polymerase. The PCR amplification was performed under the following conditions: 94 °C for 1 min followed by 30 cycles of 94 °C for 1 min, 65 °C (bacteria) or 55 °C (fungi) for 1 min and 72 °C for 1 min, followed by a final elongation at 72 °C for 10 min. Bacterial and fungal amplicons were sequenced using Illumina MiSeq (Waltham, San Diego, CA, USA) by an INRAE GeT-PLaGE platform (Castanet-Tolosan, France), generating 250 bp paired-end reads. Raw sequence data were deposited at the Sequence Read Archive of the National Center for Biotechnology Information under the BioProject number PRJEB50379.

Sequence data were processed and analyzed using the rANOMALY package [14]. Briefly, the processing of raw reads in this package is based on the dada2 R package, looking for potential sequencing errors resulting from the Illumina process, correcting these errors and defining amplicon sequence variants (ASVs) [15]. Taxonomic assignment of bacterial sequences was based on two databases, namely DAIRYdb v2.0 and SILVA 138, keeping the assignment with the highest confidence or the deepest taxonomic rank [16,17]. Fungal taxonomic assignment was based on the UNITE v8.2 Fungi database [18]. The output file was a phyloseq R object, which was used for all subsequent statistical analyses [14].

### 2.4. Lipidomics and Volatolomics

Lipidomic and volatolomic data were acquired from the study published by Frétin et al. [4]. Briefly, lipidome was determined from 100 mg of lyophilized and ground cheese at D150. After an extraction methylation procedure (methanol/boron trifluoride 95:5 *v/v*), FA were identified by gas chromatography, as described by Lerch et al. [19]. Purge-and-trap extraction was used in association with GC-MS to acquire volatolomic data on 10 mL of an ultraturraxed 10% *w/w* suspension of grated cheese in ultra-high quality water [20].

### 2.5. Statistical Analyses

Barplots were built using Microsoft Excel, only considering ASVs with relative abundance ≥1% in at least one type of sample (P- or M-derived milk, cheese core or cheese rind). Plots were drawn by type of milk (raw vs. pasteurized) and type of animal feeding (P-derived vs. M-derived milks and cheeses). All statistical analyses were performed using the rANOMALY package [14]. α-diversity indices were calculated, namely the number of observed ASVs and Chao1 for richness, and Shannon and Simpson indices for diversity. Values were compared by ANOVA, and Tukey’s HSD tests were performed for pairwise comparisons when necessary. Regarding β-diversity, Bray-Curtis dissimilarity matrices were built, and results were plotted using canonical correspondence analysis (CCA). PERMANOVA was performed to look for significant differences in microbial community structure between types of samples (i.e., cream origin, time of sampling and interaction between both factors). P-values were adjusted using the FDR method. Differential analysis was performed at the species taxonomic rank, merging results of three methods, namely DESeq2, MetaGenomeSeq and MetaCoder [21,22,23].

Before looking for potential correlations between FA/VOCs profiles and rind microbiota, a pre-selection of variables was performed as follows. Regarding microbiota, only species represented by at least five reads and at least two samples were considered. Regarding FA and VOCs, only molecules previously identified by [4] as present at significantly different concentrations between P- and M-derived cheese rinds were included in the analysis. After normalization, Pearson correlation matrices were built between metabarcoding data and conserved FA or VOCs, and the respective heatmaps were built using the R package heatmaply with default parameters for clustering, also including α-diversity indices [24]. For all analyses, *p*-values ≤ 0.05 were considered significant.

## 3. Results

### 3.1. Microbial Communities from Milk to Cantal Cheese

Bacterial and fungal communities of raw milk supplemented with either P- or M-derived pasteurized cream and of derived *Cantal* cheeses were studied using DNA metabarcoding. Regarding bacterial communities, a total of 2,279,767 reads were obtained. Overall, Firmicutes was the dominant phylum in this analysis, representing more than 85% of reads. A total of 167 ASVs was identified, belonging to 65 genera. Among these ASVs, 32 were common between all types of samples, i.e., raw milk, unripened cheeses, cheese rinds and cheese cores. Forty ASVs were only found in rinds. For all results detailed hereafter, raw milk cheeses and pasteurized milk cheeses were considered separately. Figure 2 presents cumulative barplots of relative abundances of the major bacterial genera in all types of samples.

Regarding raw milk, the dominant taxa were *Hafnia*, *Lactococcus* and *Serratia*. *Hafnia* was still present in cheeses at D3, but its relative abundance was progressively reduced during ripening. Unsurprisingly, the composition of rinds differentiated from that of cores during ripening, with the emergence of bacterial genera such as *Brevibacterium*, *Brachybacterium*, *Yaniella* and *Serratia*. The same trend was observed regarding samples manufactured from pasteurized milk. No major effect of cream origin was observed on raw milk bacterial profile at the genus rank. The influence of cream origin on other samples was limited, with the exception of raw milk cheeses at D3 and raw milk cheese rinds at D150.

Overall, 30 fungal ASVs were identified using ITS metabarcoding from a total of 834,900 sequence reads. Figure 2 presents relative abundances of major fungal genera in cheese rinds. The results detailed hereafter were observed for both pasteurized and raw milk cheeses. At D30, the dominant fungal species on *Cantal* rinds were *D. hansenii* and a *Penicillium* sp. Progressively, the relative abundances of both taxa were reduced during ripening, concomitantly with an increase in that of *S. casei*. In P-derived cheese rinds at D150, an undetermined *Microascaceae* was present at a sub-dominant level. At all sampling times, the relative abundance of *D. hansenii* was higher in P-derived than in M-derived rinds.

### 3.2. Diversity of Microbial Communities from Milk and Cantal Cheese during Ripening

Richness (number of ASVs and Chao1) and diversity (Simpson and Shannon indices) parameters were calculated to characterize α-diversity (see Appendix A). Richness and diversity were generally higher in rinds than in cores at D150 for both raw and pasteurized milk cheeses. For raw milk cheeses, the only significant effect of cream origin on bacterial communities was observed with Simpson indices for cores at D150, with greater diversity in M-derived cheeses (*p*-values = 0.033). For pasteurized milk cheese cores at D150, a significant difference in bacterial richness was observed depending on cream origin. A strong influence of cream origin on Shannon (*p*-value = 0.044) and Simpson (*p*-value = 0.048) indices were identified at D150, regarding fungal communities, with a higher diversity in P-derived cheese rinds.

Potential differences in community structure were assessed through CCA (Figure 3). As results for raw and pasteurized milks were similar, only raw milk cheese samples were displayed. Ripening time significantly impacted bacterial (Figure 3A) and fungal (Figure 3B) community structure. When considering all samples independently of ripening time, no effect of cream origin was identified (Figure 3C,D). However, interaction cream origin x ripening time had a significant influence on bacterial and fungal communities (Figure 3E,F).

### 3.3. Differentially Abundant Bacterial and Fungal Taxa between P- and M-derived Cheese Rinds

Differential analyses were performed to look for differentially abundant fungal and bacterial species between P- and M-derived cheese rinds at D30, D90 and D150. Species were considered dominant in the case of relative abundance ≥ 1%, sub-dominant when 0.1% ≤ relative abundance < 1% and rare when relative abundance was <0.1%. Figure 4 presents results of differential analyses for bacterial and fungal communities, respectively. Color intensity corresponds to the DESeqLFC value. Relative abundances of concerned taxa are detailed in Appendix A. Globally, the number of differentially abundant bacterial species between P- and M-derived cheese rinds increased during ripening but was lower and more stable in pasteurized milk cheese rinds. The number of differentially abundant fungal species was variable during ripening, independently of milk heat treatment.

In raw milk cheese rinds at D30, only two dominant species assigned to *Brevibacterium* spp. and *Enterobacter* spp. contributed to differentially defining M- and P-derived bacterial communities, respectively. At D90, P-derived cheese rinds were characterized by significantly higher abundances of *Brevibacterium* spp., *Brachybacterium* spp. and *Yaniella halotolerans*. Finally, at D150, *Brachybacterium* spp. were more abundant in P-derived rinds, while it was the case of *Lactococcus* spp. in M-derived rinds. At the same sampling time, some sub-dominant taxa were also allowed to differentiate P-derived cheese rinds, including *Nocardiopsis* sp., *Stackebrandtia nassauensis*, *Ruania aldibiflava, Corynebacterium* sp. and *Dietzia timorensis*. Fungal communities were characterized by higher levels of *D. hansenii* in P-derived rinds throughout ripening, while a higher abundance of *Penicillium* sp. was observed in M-derived cheese rinds. *Microascaceae* family and *S. casei* were characteristic of P- and M-derived rinds at D150, respectively. At a sub-dominant level, several *Mucor* species were more abundant on P-derived rinds. Conversely, an unidentified species of the genus *Yamadazima* had a higher relative abundance in M-derived rinds.

In pasteurized milk cheese rinds, P-derived samples were characterized by higher levels of *Enterobacter* spp. throughout the whole ripening process. At D150, these rinds were also characterized by higher levels of *Brachybacterium* spp. and of *Hafnia alvei*. Minor taxa, including *Flavobacterium* sp. and *Acinetobacter* sp., also contributed to defining bacterial profiles of the final P-derived rinds. M-derived rinds were characterized by higher levels of *Brevibacterium* spp. during ripening (D30 and D90). At D150, only two minor taxa of the genus *Jeotgalicoccus* characterized M-derived cheese rinds. Dominant fungal populations according to cream origin were similar to what was observed for raw milk cheese rinds. At a sub-dominant level, M-derived rinds were characterized by a higher abundance of *Yarrowia lipolytica* and *Kluyveromyces* sp.

### 3.4. Potential Correlations between Final Rind Microbial Communities and Profiles of FA and VOCs

Heatmaps of correlations between metabarcoding data and FA (Figure 5) or VOCs (Figure 6) were plotted. Both heatmaps were clearly vertically cleaved into two parts, FA and VOCs being clustered according to cream origin. Since no differences in the gross chemical composition of the cheese, pH and mineralization were associated with the origin of the cream, regardless of the treatment of the milk (Appendix A), these data were not considered in the integrative analysis. In contrast, the inclusion of α-diversity indices for bacterial and fungal communities within the matrices allowed a more global interpretation of the results. Regarding bacteria, Simpson and Shannon indices were correlated with the concentration of C18:1 *cis13* (*p*-values = 0.008 and 0.010, respectively). Concentrations of 2-propanol, 2-pentanol, 2-heptanol and methylhexanoate were correlated with bacterial richness (all *p*-values < 0.050), whereas concentrations of 1-pentanol, 2-pentanol, 2-heptanol and 3-methyl-2-pentanol were impacted by bacterial diversity (*p*-values < 0.05). While focusing more on bacterial species, the concentration of C18:1 *cis13* was correlated with *Brachybacterium* spp. (*p*-value = 0.042) and *Nocardiopsis* sp. (*p*-value = 0.029), the latter one also being correlated with CLA concentration (*p*-value = 0.044). No significant correlations were identified between species significantly more abundant in M-derived cheese rinds, i.e., *Lactococcus* spp., *Y. halotolerans*, *S. equorum* and *Lactobacillus* sp., and FA concentrations. Regarding VOCs, concentrations of 1-pentanol and 3-methyl-2-pentanol (all *p*-values < 0.050) were correlated with most species identified as more abundant in P-derived cheese rinds. Among species more abundant in M-derived cheese rinds, *Lactococcus* spp. were significantly correlated with 2-propanone (*p*-value = 0.042), while *Lactobacillus* sp. was correlated with methylbutanoate, methylhexanoate, 2-propanol, 2-pentanol and 2-heptanol (all *p*-values < 0.050).

The number of fungal ASVs as well as Shannon and Simpson indices were clearly correlated with C18 FA (C18:0, C18:1 *cis*9, C18:1 *cis13*, C18:2n-6, C18:3n-3 and CLA) and negatively correlated with both medium-chain saturated and unsaturated FA, including C10:0, C12:0, C14:0 and C16:0. Concentrations of 1-pentanol, 3-methyl-2-pentanol and methylbenzene were correlated to Shannon and Simpson indices. Regarding unidentified *Microascaceae*, *D. hansenii* and *Mucor circinelloides*, correlations similar to what we reported hereabove for Simpson and Shannon indices were revealed (all *p*-values < 0.050). In M-derived cheese rinds, *S. casei* was only correlated with concentrations of iso C16 (*p*-value = 0.046), but highly negatively correlated with C18:1 *cis*13 (correlation coefficient = −0.91; *p*-value < 0.001). Generally, concentrations of 1-pentanol, 3-methyl-2-pentanol and methylbenzene were correlated with species more abundant in P-derived rinds. Regarding VOCs, a significant negative correlation was observed between relative abundance of *S. casei* and concentrations of 1-pentanol (*p*-value = 0.016) and 3-methyl-2-pentanol (*p*-value = 0.004).

## 4. Discussion

Cheese is a complex matrix whose biochemical and microbial composition is influenced by the conditions in which it is produced. Cow feeding is one of these factors, as it influences milk and cheese FA and VOC profiles [2,6,25]. In this study, we integrated multiomic data to investigate the influence of cream origin on bacterial and fungal communities of *Cantal* cheeses. Firstly, DNA barcoding was used to characterize microbial communities based on ITS and 16S sequences for fungi and bacteria, respectively. Various ripening times were considered, including cheese cores at D3 and D150 and cheese rinds at D30, D90 and D150. Metabarcoding data were related to lipidomics and volatolomics data previously acquired from the same cheese samples [4].

Cheese rinds, influenced by the development and dynamics of fungal and bacterial communities, contribute intensively to cheese sensorial properties [11]. Although remaining dominant in rinds, the relative abundance of *Lactococcus* progressively decreased during ripening, simultaneously with the increasing proportion of *Brevibacterium* and *Brachybacterium*. Both genera include species that were used as ripening starters, namely *B. linens* and *B. tyrofermentans*, and are typically found in cheese rinds [1].

Some influence of cream origin on the cheese-dominant bacterial community was identified. Overall, ripening bacteria, including *Brevibacterium* spp. and *Brachybacterium* spp., had a significantly higher relative abundance in P-derived cheeses, while M-derived cheeses promoted the maintenance of higher *Lactococcus* levels. P-derived raw milk cheese rinds were also characterized by higher abundances of some sub-dominant taxa in comparison to M-derived cheeses rinds. Among these, *Nocardiopsis* spp., a species commonly observed on cheese rinds since the emergence of NGS, was already identified as prevalent in pasture-derived *Cantal* cheeses [2]. *Corynebacterium* sp., *Dietzia psychralcaliphila* and *Dietzia timorensis* were also more abundant in P-derived cheese rinds at D150. The presence of *Dietzia* spp. was already mentioned in milk primary production environment [2] and in final cheese rinds [26]. *Corynebacterium and Dietzia* are responsible for the production of carotenoids [26]. They could thus be associated with the yellower color observed during sensorial analyses on P-derived cheeses [4].

The dynamics of fungal communities during *Cantal* cheese ripening had never been investigated before. Overall, 32 ASVs were identified in *Cantal* cheese rinds. This number ranks among the highest values reported in previous works on uncooked pressed cheeses, in which between 10 and 30 OTUs were identified [8,10]. Globally, rind fungal profiles intensively evolved during ripening. At D30, *D. hansenii* was the most abundant taxon in both P- and M-derived cheese rinds, followed by a *Penicillium* sp., *D. hansenii* and *P. fuscoglaucom* were added as ripening starters for *Cantal* cheese manufacturing [2]. *D. hansenii* was reported as the dominant fungal species in several cheese rinds, including *Saint-Nectaire*, *Maroilles*, *Munster*, *Raclette* and *Gruyère* [27]. The third fungal starter, *S. casei*, had a limited relative abundance at D30 (i.e., 3–5%), but its proportion was intensively increased at D90 (67%) and D150 (75–93%). This tendency was similar between raw and pasteurized milk cheese rinds. The presence of *Scopulariopsis brevicaulis* as a dominant species was already mentioned in semi-hard cheese from Austria and Denmark [11,28]. At first glance, this taxon was not identified in *Cantal* using ITS DNA metabarcoding. Nevertheless, between 7 and 10% of sequence reads of raw and pasteurized P-derived cheese rinds were assigned to members of the *Microascaceae* family, the latter one including *Scopulariopsis* spp. Using NCBI Nucleotide BLAST [29], this ASV was assimilated to *S. brevicaulis* with sequence identity and coverage of 100%. Interestingly, cream origin had a great influence on the development of this species in cheese. The presence of organisms less frequently mentioned on cheese rinds was observed during the present work, including *Yamadazima* spp. or *Mucor* spp. *Yamadazima* spp. are highly halotolerant yeasts already isolated from dairies and cheese-processing environments, including cheese brines [27]. *Mucor* spp. are molds described as responsible for defects in cheese, including unexpected flavors and colors, but some species, like *Mucor fuscus*, could be desired in some cheeses for their technological role through their proteolytic and lipolytic activities [27,30]. *Mucor* spp. and *Yamadazima* spp. were differentially abundant between P- and M-derived cheeses.

The main hypothesis of this work was that cream origin, characterized by variations in cheese rind lipidomic profiles, is responsible for modulation of cheese bacterial and fungal communities, themselves associated with differential production of VOCs through their metabolic activities. Thus, we decided to look for correlations between microbiota and FA profile, with a focus on differentially abundant species between P- and M-derived cheese rinds. Similarly, the influence of these species on VOC profiles was also investigated. Figure 5 revealed a strong influence of cheese rind FA profile on microbial communities, especially fungi. Previously, it was observed that concentrations of C18 FA were higher in P-derived cheese rinds, while those of short- and medium-chain FA were higher in M-derived cheese rinds [4]. Although their influence on bacteria seemed limited, monounsaturated FA (MUFA) and PUFA could play a potential role in the differentiation of fungal communities between P- and M-derived cheese rinds. In particular, high concentrations of MUFA and PUFA could have a negative effect on the development of the yeast *S. casei*. This species is commonly added as a ripening starter during *Cantal* cheese manufacture and contributes to the typical ochre color of rinds. During a sensorial evaluation, trained panelists attributed significantly lower notes to P-derived rind spot salience and quantity [4]. P-derived cheese rinds were also thinner than M-derived ones. Potentially, all these observations could be associated with the lower abundance of *S. casei*, identified using DNA metabarcoding and significantly negatively correlated with C18:1 cis9. A phenomenon of pronounced oiling-off was already described as inhibitive on yeasts, e.g., *Mucor* sp. in *Saint-Nectaire* cheese [19], but it was not the case in the present study. Indeed, *Mucor* sp. had a significantly higher relative abundance in P-derived than in M-derived cheese rinds. Contrary to *S. casei*, growth of *S. brevicaulis* was promoted in cheese rinds rich in MUFA and PUFA, resulting in a final relative abundance of nearly 10%. Nevertheless, the link between *S. brevicaulis* and FA profiles has never been described, and it should always be reminded that a correlation does not necessarily involve a causal relationship.

Panelists who tasted P- and M-derived *Cantal* cheeses did not identify significant differences in aromas and flavor [4]. Nevertheless, chemical analyses revealed significant differences in the concentration of some alcohols, ketones, esters and aldehydes. Some of them derived from FA, including butanal, pentanal, heptanal, 2-butanone and 2-propanol [4]. As a consequence, cream origin could have an influence on the final concentrations of these VOCs. Among these molecules, ketones and related alcohols like 2-pentanol and 2-heptanol were more concentrated in M-derived cheese rinds, while aldehydes had higher concentrations in P-derived cheese rinds. From Figure 6, several microbial species-VOC pairs were significantly correlated. In studies published by [31,32] on *Historic Rebel* and *Pélardon* cheeses, a strong correlation was observed between *Lactococcus* spp. and ketones. It was also the case in the present study, and *Lactococcus* spp. were significantly more abundant in M-derived cheeses, in which final concentrations of ketonic VOCs were higher. Similarly, our work and previous studies converge on showing that the concentration of secondary alcohols was correlated to the presence of *Lactobacillus* spp. [31,32].

In *Pélardon*, *D. hansenii* was not correlated with particular VOCs and was thus considered as a poorly aromatic strain. In the present work, *D. hansenii* and other *Debaryomyces* spp. were more abundant in P-derived rinds and were correlated with concentrations of methylbenzene. The presence of *S. brevicaulis* was identified at a dominant level after the extended ripening of *Pélardon* cheese [32]. This relative abundance was negatively correlated to the concentration of ketonic VOCs. Similar tendencies were observed in *Cantal* cheese rinds. In these cheeses, a significant positive correlation was also identified between *S. brevicaulis* and concentrations of aldehydes, methylbenzene and 1-pentanol. *S. brevicaulis* is a mold known for its intense proteolytic activity and should potentially be considered as a cheese spoiler when found in the dominant position [32]. Nevertheless, at the extent of P-derived *Cantal* cheese rinds (i.e., a relative abundance < 10%), no perceptible deleterious effects on cheese sensorial properties were reported by the trained panelists [4].

In addition to the already observed correlations, the present study identified novel microbial species-VOC correlations. For instance, a lot of bacterial species that were more abundant in P-derived rinds, including *Corynebacterium* sp. and *Dietzia* spp., were correlated to 1-pentanol. Interestingly, this VOC was strongly negatively correlated to levels of *S. casei*, suggesting that P-derived cheese lipidome and volatolome could create a less favorable environment for the optimal growth of this yeast.

It is difficult to certify that VOCs correlated with given taxa effectively result from the metabolic activities of these microorganisms without performing specific investigations. Furthermore, if the volatolomic profile is effectively influenced by cheese microbiota, it can also be associated with the lipidomic profile. The inhibitory activity of some VOCs on specific microbial species also adds complexity in the understanding of the relationships between microbial, FA and VOC profiles. As an example, VOCs produced by the secondary metabolism of *G. geotrichum* could inhibit the growth of *Brachybacterium* spp. and promote the growth of *Vibrio* spp. and *Psychrobacter* spp. [33]. Finally, it is important to mention that the identified VOCs were not necessarily produced by surface microbiota. Indeed, due to VOC volatility, molecules liberated in the cheese core could diffuse to the upper layers.

## 5. Conclusions

Several studies have already used multiomic approaches to relate cheese microbial, FA and/or VOC profiles but, to our knowledge, the present work is the first to focus on the influence of milk FA composition that derived from pasture-based (P) or maize silage-based (M) animal diet, on all these parameters. Additionally, *Cantal* cheese rind microbial communities were characterized for the first time by using NGS technologies. We showed that, in addition to a different cheese lipidome, cream origin modulates cheese microbiota and especially fungal communities. Similarly, correlations can be established between cheese VOC profiles and microbiota. Nevertheless, a comprehensive understanding of interactions among all these factors remains a challenge and requires more specific investigations. In particular, the differential relative abundance of several fungal taxa such as *S. casei* between P-derived and M-derived cheese rinds, regardless of the milk treatment (pasteurized or raw), is a topic of prior interest. Indeed, this yeast has been historically added as a starter for *Cantal* cheese ripening and contributes to the expected typical rind aspect. These results raised questions about the tripartite match between milk primary production conditions, expected cheese rind appearance and characteristics of added ripening starters. Thus, it would be worthwhile in future work to evaluate the effects of fat composition on different strains of the species used as starter cultures. In addition to the fatty acid composition, the concentration of certain molecules, including carotenes, vitamins A and E, is known to be influenced by animal diet and may vary according to the season. Their potential influence on cheese microbiota could deserve more attention.

## Figures and Tables

**Figure 1 microorganisms-10-00334-f001:**
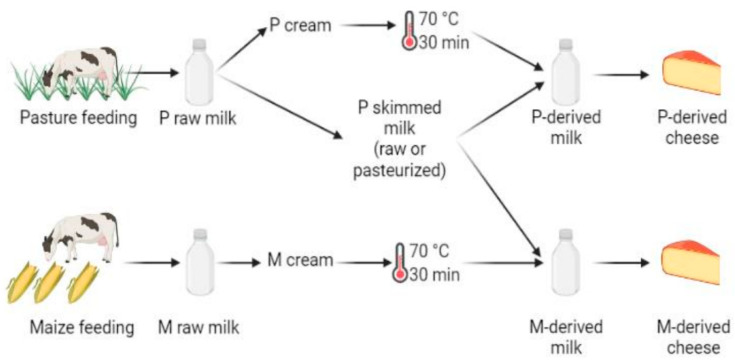
Experimental design, performed in triplicate. The following samples were used for DNA metabarcoding analyses: P- and M-derived raw milk, unripened P- and M-derived cheese (i.e., day (D)3), P- and M-derived cheese rinds at D30 and D90 of ripening, P- and M-derived cheese rinds at the end of ripening (i.e., D150) and P- and M-derived cheese cores at D150 (figure created using BioRender.com, accessed on 4 November 2021)).

**Figure 2 microorganisms-10-00334-f002:**
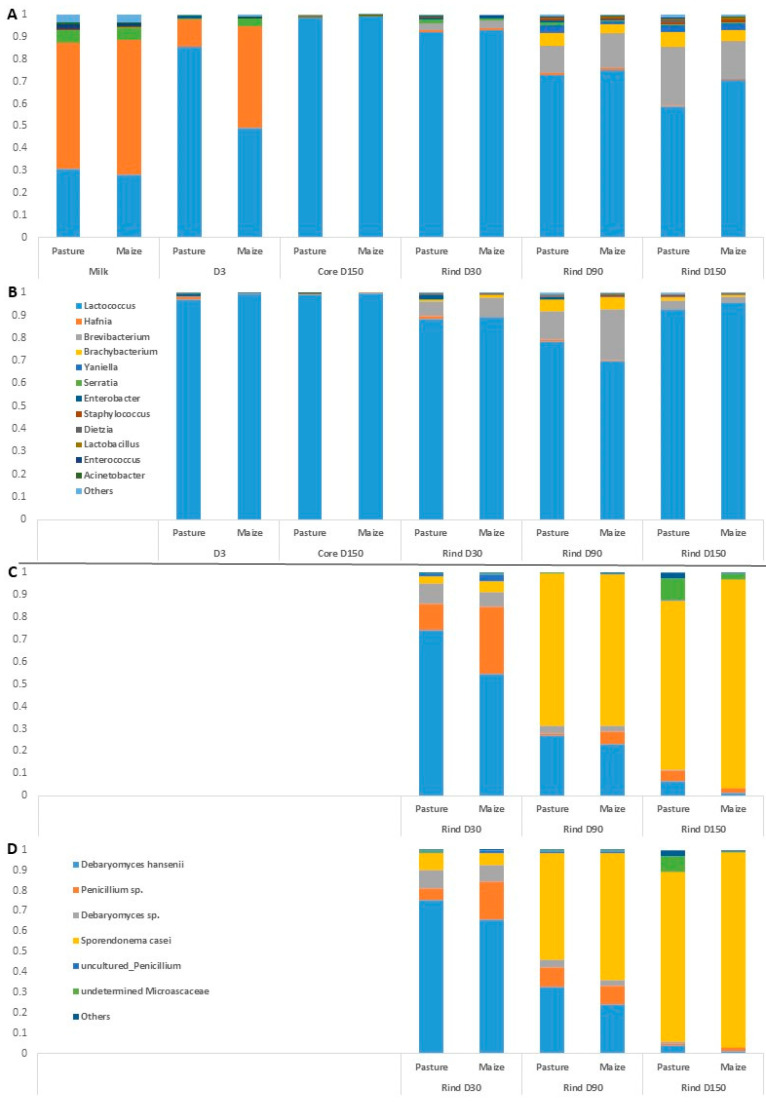
Average relative abundances (*n* = 3) of the major bacterial (**A**,**B**) and fungal (**C**,**D**) taxa in milks and cheeses according to cream origin. (**A**,**C**) Cantal manufactured from raw milk; (**B**,**D**) Cantal manufactured from pasteurized milk; only genera with relative abundances >1% in at least one type of sample are plotted.

**Figure 3 microorganisms-10-00334-f003:**
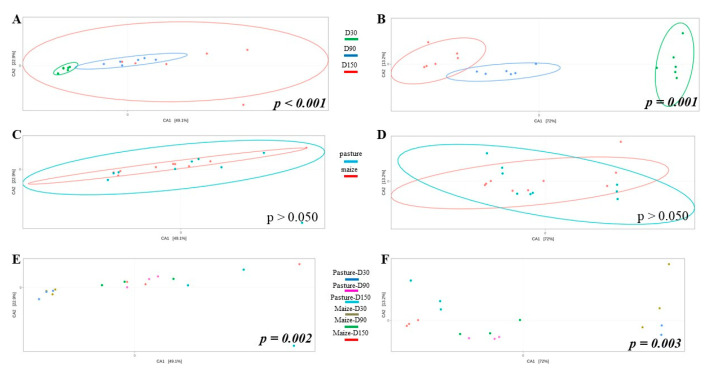
CCA built from Bray-Curtis dissimilarity matrix and comparing bacterial (**A**,**C**,**E**) and fungal (**B**,**D**,**F**) community structure of raw milk cheese rinds as function of ripening time (**A**,**B**), cream origin (**C**,**D**) and interaction between both factors (**E**,**F**); *p*-values written in italic bold are significant.

**Figure 4 microorganisms-10-00334-f004:**
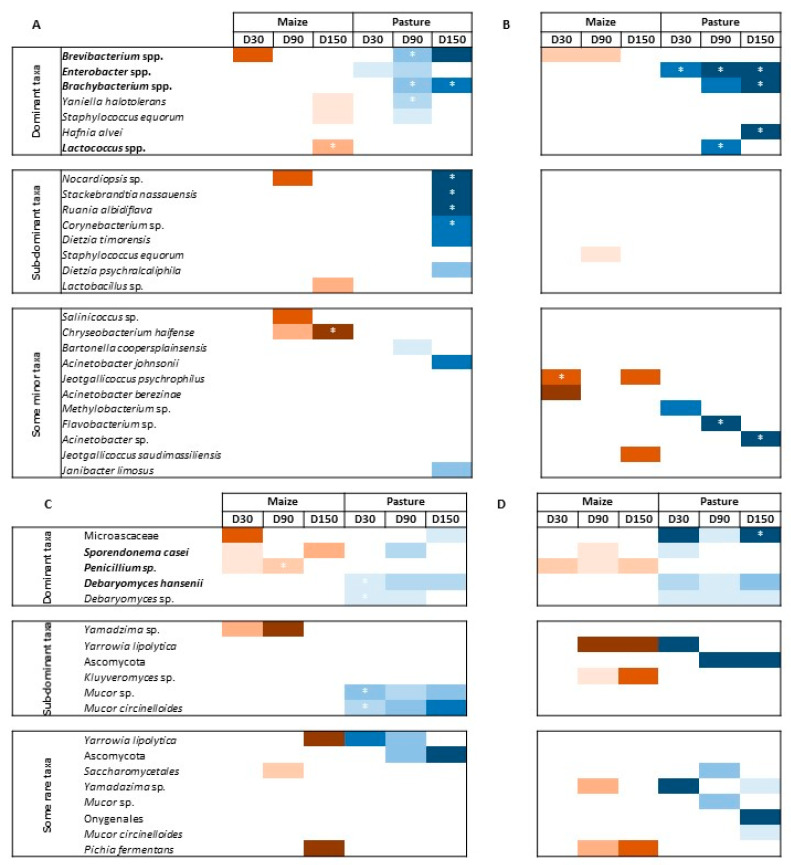
Differential analysis for bacterial (**A**,**B**) and fungal (**C**,**D**) communities between M- and P-derived cheese rinds during ripening; (**A**,**C**), raw milk cheese rinds; (**B**–**D**), pasteurized milk cheese rinds; color intensity depends on DESeqLFC value; * means that abundance of the taxon is significantly different (i.e., corrected *p*-value ≤ 0.05); taxa written in bold are likely to be associated with starter cultures; dominant taxa have a relative abundance ≥ 1%; sub-dominant taxa have a relative abundance between 0.1 and 1%; minor taxa have a relative abundance < 0.1%.

**Figure 5 microorganisms-10-00334-f005:**
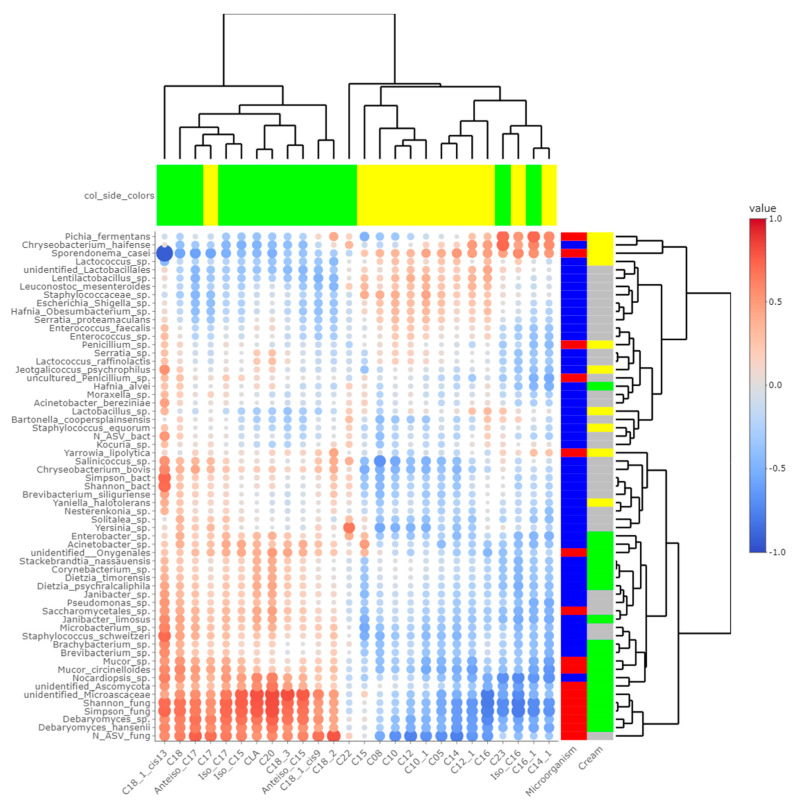
Correlations between metabarcoding data and fatty acids identified by [4] as significantly different between P- and M-derived cheese rinds; blue to red scale indicates the value of correlation coefficient; point size is proportional to –log_10_ of p-value; colors in column margin indicate whether the concentration of the concerned FA was significantly higher in P- (green) or M- (yellow) derived cheese rinds; colors in line margin indicate whether the concerned taxon is a bacterium (blue) or fungi (red) and whether taxon was more abundant in P- (green) or M- (yellow) derived cheese rinds or with a similar abundance between both rinds (grey).

**Figure 6 microorganisms-10-00334-f006:**
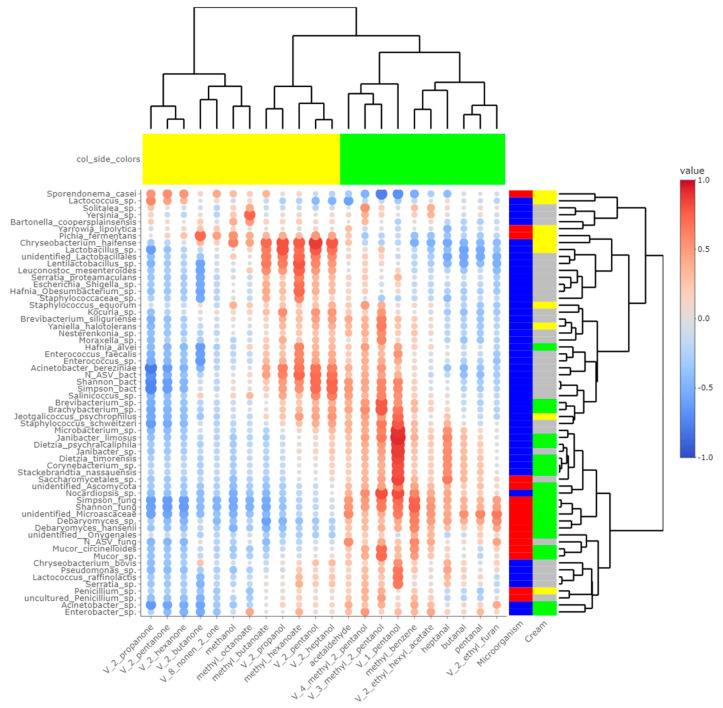
Correlations between metabarcoding data and volatiles identified by [4] as significantly different between P- and M- derived cheese rinds; blue to red scale indicates the value of correlation coefficient; point size is proportional to –log_10_ of *p*-value; colors in column margin indicate whether concentration of the concerned FA was significantly higher in P- (green) or M- (yellow) derived cheese rinds; colors in line margin indicate whether the concerned taxon is a bacterium (blue) or fungi (red) and whether taxon was more abundant in P- (green) or M- (yellow) derived cheese rinds or with a similar abundance between both rinds (grey).

## Data Availability

Raw sequence data were deposited at the Sequence Read Archive of the National Center for Biotechnology Information under the BioProject number PRJEB50379.

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
