# Peer review of "Integration of Multiomic Data to Characterize the Influence of Milk Fat Composition on Cantal-Type Cheese Microbiota"

_microorganisms, 2022, doi:10.3390/microorganisms10020334_

Round 1
Reviewer 1 Report
The aim of the research falls within the thematic scope of the journal.
The purpose of this study was to investigate the influence of milk fatty acid composition on the dynamics of bacterial and fungal communities in Cantal-type cheeses during ripening, using metabarcoding.
The topic seems to be very interesting. The research was prepared with the use of modern analytical methods and techniques as well as data analysis methods.
The manuscript is well prepared, requires only some adjustments. Most of the comments concern incorrect, in my opinion, the way of citing (literature references without the names of the authors) (lines 73, 86, 91, 92, 108, 130, 156, 175, 178, 386, 401, 435, 443, 467, 489, 493, 498, 500).
There is no literature reference on line 481 ("Panelists who tasted P- and M-derived Cantal cheeses did not identify significant differences in aromas and flavor [2].").
All remarks are marked in the text of manuscript in the review mode.
After these minor corrections have been made, the article can be referred for further editorial work.

Author Response
R1: The manuscript is well prepared, requires only some adjustments. Most of the comments concern incorrect, in my opinion, the way of citing (literature references without the names of the authors) (lines 73, 86, 91, 92, 108, 130, 156, 175, 178, 386, 401, 435, 443, 467, 489, 493, 498, 500).
AU: The way of citing concerned references was addressed throughout the manuscript.
R1: There is no literature reference on line 481 ("Panelists who tasted P- and M-derived Cantal cheeses did not identify significant differences in aromas and flavor [4].").
AU: The reference [4] has been added to this sentence.
R1: All remarks are marked in the text of manuscript in the review mode.
AU: All remarks directly suggested in the text of the manuscript have been addressed.
Reviewer 2 Report
The Manuscript sent to me for review entitled “Integration of multiomic data to characterize the influence of milk fat composition on Cantal-type cheese microbiota” is interesting and presents an innovative approach. The research was well planned, the concept is interesting and it should be emphasized that the Authors used current and desired methods (NGS). The research focuses on the influence of the milk fatty acid (FA) composition on changes in bacterial and fungal compositions in the core and rind of Cantal cheese during ripening. The originality of the research, careful planning of the research, a large amount of work by the Authors as well as an extensive statistical analysis should be emphasized.
Despite the overall very good impression the Manuscript needs to be improved. My suggestions are as follows:
Introduction:
The Introduction section is too detailed. The line 61-94 in particular presents the current knowledge in too much detail. I suggest rearranging this fragment to indicate more precisely the desirability/merits of Your research presented in the current Manuscript. Some of these fragments could be included in the Discussion.
It should be emphasized that the Authors clearly presented the purpose of the research, which I greatly appreciate.
Materials and Methods:
What did the Authors suggest in the selection of cows/groups? Why were Holstein and Montbéliarde cows selected? What was the dictation of such a size and diversity of groups?
Maybe it would also be good in the Materials and Methods section to provide an explanation of the abbreviation P and M?
Figure 1: Please explain the meaning of the letter “D” in relation to D3, D30, D90 etc. in the description of the Figure.
Line 197-204: Why were only so characterized samples taken into account in relation to the microbiota and in relation to FA and VOCs?
Why are D3, D30, D90 and D150 selected for the sampling times? According to the Authors' knowledge, what changes is this related to?
Results:
The graphical presentation of the results does not raise any objections. The description of the results is understandable but it focuses on all the results, not the most important insights. Descriptions are too detailed. The Authors describe all the results, not just those relevant to the essence of the Manuscript. Listing all the results is distracting and makes it difficult to focus on the most important insights. I suggest shortening and presenting a logical sequence.
In some places the Authors use “sp.” and sometimes “spp.” for the names of microorganisms. Given the differences, is such use by Authors appropriate?
Is it good that the Authors relate the current results to the previously obtained lipidomic and volatolomic data? How does this relate to the repeatability of previous results? This is not a complaint, but just a question.
The results lack dependencies / differences depending on the starter cultures used.
Discussion:
The Discussion is too long. The Authors describe their results here too extensively. Please shorten and compare the MOST IMPORTANT results with the current knowledge.
Paragraph fom Line 453 the Discussion looks good. This was the essence of Your research, and this is what the goal shows. A broad description of the results is unnecessary.
Author Response
Introduction:
R2: The Introduction section is too detailed. The line 61-94 in particular presents the current knowledge in too much detail. I suggest rearranging this fragment to indicate more precisely the desirability/merits of Your research presented in the current Manuscript. Some of these fragments could be included in the Discussion.
AU: As suggested, the authors removed unnecessary details from lines 61 to 94 of the manuscript.
Materials and Methods:
R2: What did the Authors suggest in the selection of cows/groups? Why were Holstein and Montbéliarde cows selected? What was the dictation of such a size and diversity of groups?
AU: Cow groups were designed to be representative of the main breeds found in farms from the concerned geographical area (Auvergne Region). The animals were divided into these 2 groups based on breed, parity (41 and 42% of primiparous cows in P and M groups, respectively) and lactation stage (228 ± 121 and 223 ± 59 DIM in P and M groups, respectively), so that the characteristics of the animals in the two groups were similar despite the diversity within groups. The number of animals in each group was dictated by the amount of milk and cream necessary for cheese manufacture.
Precisions regarding these groups have been added to the manuscript.
R2: Maybe it would also be good in the Materials and Methods section to provide an explanation of the abbreviation P and M?
AU: As suggested, P and M abbreviations have been described in the Materials and Methods section.
R2: Figure 1: Please explain the meaning of the letter “D” in relation to D3, D30, D90 etc. in the description of the Figure.
AU: Letter D is used as an abbreviation for the word ‘day’. An explanation has been added to the legend of Figure 1.
R2: Line 197-204: Why were only so characterized samples taken into account in relation to the microbiota and in relation to FA and VOCs?
AU: Taxa represented by a low amount of sequences and present in only one sample could result from errors occurring during DNA sequencing or during in silico analysis of sequence reads. Consequently, considering all these taxa during subsequent analyses could result in the identification of artifactitious correlations and to erroneous conclusions. Regarding FA and VOCs, a previous study performed on the same cheese samples already identified molecules with significantly different concentrations between P- and M-derived samples. It was thus decided to consider only these FA and VOCs for the integrative analysis.
R2: Why are D3, D30, D90 and D150 selected for the sampling times? According to the Authors' knowledge, what changes is this related to?
AU: The choice of sampling times was based on expert knowledge of the cheese technology studied, in particular on the evolution of the surface aspect of cheeses during ripening, and on previous studies on cheeses with a similar production process such as Salers cheese also sampled at D30, D90 and D150 during ripening (https://doi.org/10.3168/jds.S0022-0302(05)73069-1).
D3 corresponds to the first day of ripening. At D30, ripening microbiota only starts its development. D90 corresponds to an intermediate point in the development of the expected cheese rind. Finally, D150 corresponds to an average ripening duration for Cantal cheese.
Details about the rationale for the sampling dates have been added to the manuscript.
Results:
R2: The graphical presentation of the results does not raise any objections. The description of the results is understandable but it focuses on all the results, not the most important insights. Descriptions are too detailed. The Authors describe all the results, not just those relevant to the essence of the Manuscript. Listing all the results is distracting and makes it difficult to focus on the most important insights. I suggest shortening and presenting a logical sequence.
AU: Results section was summarized, focusing only on major differences between P- and M-derived samples for raw or pasteurized milk cheeses.
R2: In some places the Authors use “sp.” and sometimes “spp.” for the names of microorganisms. Given the differences, is such use by Authors appropriate?
AU: All « sp. » and « spp. » were checked an addressed when necessary. « sp. » is used when a single species is concerned, while « spp. » is used when several species of a given genus are concerned by the sentence.
R2: Is it good that the Authors relate the current results to the previously obtained lipidomic and volatolomic data? How does this relate to the repeatability of previous results? This is not a complaint, but just a question.
AU: In fact, cheese samples considered by the present study are exactly the same as those on which [4] was based. Consequently, there are no bias in the analysis. As this was not clear, precisions were added to the manuscript (Discussion section, line 400) in order to avoid misunderstandings.
R2: The results lack dependencies / differences depending on the starter cultures used.
AU: In this study, all cheese samples were manufactured using the same batches of commercial starter cultures, as now specified in the manuscript. Consequently, the effect of varying starter cultures was not assessed during the present work. The authors agree it would be worthwhile in future work to evaluate the effects of fat composition on different strains of the species used as starter cultures as now mentioned as perspective of the present work.
Discussion:
R2: The Discussion is too long. The Authors describe their results here too extensively. Please shorten and compare the MOST IMPORTANT results with the current knowledge.
AU: As suggested, discussion was summarized and reorganized.
Reviewer 3 Report
Differences in the sensory properties of cheese and other dairy products produced using grass-fed cows vs other feeding methods e.g. maize is an important research area e.g. some Asian consumers can identify a barn off odour from milk produced from grass-fed cows. The use of data from multiomics along with GC-MS studies to study the influence of milk fat origin on the microbiota of Cantal-type cheese is interesting and with some modifications to the text, I am content to recommend publication.
Matters for the authors to correct/address.
- The meaning of the last sentence of the abstract is unclear and this must be rewritten.
- The description of cheese making is inadequate. The authors reference earlier work which in turn cites Martin et al (2009). To have to access two references to see what the authors did is not good practice. The authors should at least state the concentration of 0.05g/100kg DVI/DVS culture used.
Some lactococci have significant lipolytic activity. Was any characterisation of the starter done? Was the same batch of DVI/DVS culture used throughout the trials? If not, how did you control for differences in species and strain composition between trials? And yes, starter manufacturers often vary species and strains between batches! They control starter activity generally very well.
Was the milk standardised for cheesemaking and if not why not?
- The authors rightly state “Cheese is a complex matrix whose biochemical and microbial composition is influenced by conditions in which it is produced.” However, no data on the gross composition of the milk or cheese is provided. This must be provided to assess if there are differences between the cheeses.
- The cheesemaking was done over a small time period. This does not enable seasonal and other effects to be assessed. The authors at least need to mention this significant limitation.
- The authors have mentioned the effects of heat treatment on the microbiome of cheese in the introduction. Some further discussion of their findings vs raw and pasteurised milk would be helpful as this is an area of interest.
- I enjoyed reading your paper.
Author Response
R3: The meaning of the last sentence of the abstract is unclear and this must be rewritten.
AU: The sentence has been reworded and focused to highlight the novel interactions between milk fat composition and the development of fungal communities in cheeses.
R3: The description of cheese making is inadequate. The authors reference earlier work which in turn cites Martin et al (2009). To have to access two references to see what the authors did is not good practice. The authors should at least state the concentration of 0.05g/100kg DVI/DVS culture used.
AU: Key information regarding cheese manufacture have been added to the manuscript.
R3: Some lactococci have significant lipolytic activity. Was any characterisation of the starter done? Was the same batch of DVI/DVS culture used throughout the trials? If not, how did you control for differences in species and strain composition between trials? And yes, starter manufacturers often vary species and strains between batches! They control starter activity generally very well.
AU: The authors totally agree with this remark. For this reason, to avoid bias potentially associated with strain rotations between starter culture batches, all cheese batches were manufactured using the same batch of commercial starter cultures. This information was added to the manuscript.
R3: Was the milk standardised for cheesemaking and if not why not?
AU: Skimmed milk was supplemented either with pasture (P)- or maize (M)-derived pasteurized cream, to obtain the same final concentration of fat (39 g/L) in each vat. This information was added to the manuscript.
R3: The authors rightly state “Cheese is a complex matrix whose biochemical and microbial composition is influenced by conditions in which it is produced.” However, no data on the gross composition of the milk or cheese is provided. This must be provided to assess if there are differences between the cheeses.
AU: Table S5 displaying cheese gross physicochemical parameters was added as supplementary data. Since no differences in the gross chemical composition of the cheese, pH and mineralization, were associated with the origin of the cream, regardless of the treatment of the milk (raw or pasteurized), these data were not considered in the integrative analysis. This was specified in the Results section of the manuscript.
R3: The cheesemaking was done over a small time period. This does not enable seasonal and other effects to be assessed. The authors at least need to mention this significant limitation.
AU: The authors agree that the physico-chemical and microbiological characteristics of milk and cheese can vary with the season together with potential confounding factors such as animal housing and lactation stage. In the present study, to limit the number of extrinsic factors, apart from animal feeding, influencing FA composition and microbiota, we deliberately set up an experiment based on two groups of animals conducted simultaneously over a short period of time during a unique season. The need to evaluate the effects of the season in future projects was highlighted in the conclusion.
R3: The authors have mentioned the effects of heat treatment on the microbiome of cheese in the introduction. Some further discussion of their findings vs raw and pasteurised milk would be helpful as this is an area of interest.
AU: The raw vs. pasteurized comparison is indeed an area of interest but our experimental setup was primarily designed to test the effect of fat on the same milk. Due to technical constraints, it was not possible to test all modalities on the same day, so raw and pasteurised cheeses were not produced on the same day. This point has been clarified in the Materials and Methods section of the manuscript.
Nevertheless, fat composition had similar effects on microbial communities, especially fungi, in terms of diversity and composition, regardless of the heat treatment of the milk, as reported in different paragraphs of the Results section.
The Conclusion has been amended to highlight that the differential relative abundance of several fungal taxa such as S. casei, a yeast added as starter for Cantal cheese ripening, between P-derived and M-derived cheese rinds, was consistent regardless of the milk treatment (pasteurized or raw).
R3: I enjoyed reading your paper.
AU: The authors would like to thank the reviewers for all their relevant and constructive comments.
Round 2
Reviewer 2 Report
Many thanks to the Authors for comprehensive and satisfactory answers.
I recommend publishing the Manuscript.
Congratulations on Your very good research!
Reviewer 3 Report
I am content with the modified manuscript.